# *Kabuli* and *Apulian black* Chickpea Milling By-Products as Innovative Ingredients to Provide High Levels of Dietary Fibre and Bioactive Compounds in Gluten-Free Fresh Pasta

**DOI:** 10.3390/molecules26154442

**Published:** 2021-07-23

**Authors:** Michela Costantini, Carmine Summo, Michele Faccia, Francesco Caponio, Antonella Pasqualone

**Affiliations:** Department of Soil, Plant and Food Science (DISSPA), University of Bari Aldo Moro, Via Amendola 165/A, I-70126 Bari, Italy; michela.costantini@uniba.it (M.C.); carmine.summo@uniba.it (C.S.); michele.faccia@uniba.it (M.F.); francesco.caponio@uniba.it (F.C.)

**Keywords:** fresh pasta, gluten-free, *Apulian black* chickpea, high fibre, anti-nutritional compounds, bioactive compounds, texture, cooking performance

## Abstract

Gluten-free (GF) products, including pasta, are often characterised by nutritional deficiencies, such as scarce dietary fibre and excess of calories. Chickpea flour is increasingly being used by the food industries. Hulls, rich in dietary fibre and bioactive compounds, are discarded after milling. The aim of this work was to evaluate the quality features of short-cut GF fresh pasta added of hull (8% *w*/*w*) derived from *kabuli* (KH) or *Apulian black* (ABH) chickpeas, in comparison with control GF pasta prepared without hull. The enriched pasta, which could be labelled as “high fibre”, was characterised by a higher level of bioactive compounds and antioxidant activity than the control. ABH-enriched pasta showed the highest anthocyanins (33.37 ± 1.20 and 20.59 ± 0.11 mg/kg of cyanidin-3-*O*-glucoside on dry matter in raw and cooked pasta, respectively). Hull addition increased colour intensity and structural quality of GF pasta: ABH-enriched pasta had the lowest cooking loss and the highest water absorption capacity; KH-enriched pasta showed the highest firmness. No significant differences in sensory liking were found among the samples, except for “aftertaste”. Chickpea hull can be used as an innovative ingredient to produce potentially functional GF pasta, meeting the dietary needs of consumers without affecting quality.

## 1. Introduction

Traditionally made from durum wheat semolina, pasta is very appreciated due to its palatability, low cost, and ease of preparation [1]. Based on the moisture content, pasta is classified into two types: “dry” and “fresh”, with a maximum and minimum value of 12.5% and 24%, respectively [2]. Fresh pasta is growing continuously on the market due to the consumers’ unconscious belief in the close relationship between freshness and artisanal production [3]. Moreover, fresh pasta is particularly suitable for making traditional types, linked to certain geographical areas, which are highly appreciated by local consumers and tourists. In the Apulia region (Southern Italy), one of the most traditional fresh pasta shapes is called “*orecchiette*” (literally “little ears”) [4].

The increase in the incidence of celiac disease has resulted in a greater demand for GF products, including pasta. To obtain GF products with acceptable sensory properties, it is necessary to use hydrocolloids, such as guar gum, xanthan gum, carrageenan, and hydroxypropyl methylcellulose (HPMC), able to replace gluten [5]. However, these products often do not meet the nutritional needs of celiac consumers in terms of dietary fibre [6]. Therefore, in recent years, several studies have been carried out to fortify GF fresh pasta, proposing the addition of chickpea flour, fenugreek and tiger nuts [7], spirulina [8], and defatted almond flour [9].

Chickpeas (*Cicer arietinum* L.) play an important role in a healthy and sustainable diet. Chickpeas are classified into two main groups: *kabuli*, with a smooth coat characterised by large, beige-coloured seeds, and *desi*, with small, rough seeds and a black or brown coat [10]. In recent years, another type of black chickpea has been characterised both genetically [11] and nutritionally [10], the *Apulian black* chickpea (from the Apulia region). *Apulian black* chickpea is at risk of genetic erosion, but could be effectively used as ingredient to improve the nutritional characteristics of foods due to its high content of antioxidant compounds, such as anthocyanins and carotenoids [12]. This type of chickpea has been proposed to reformulate various foods, such as bakery products [12], canned purée [13], and dry pasta [14], generally after milling.

Like other plant food processing, chickpea flour production generates waste and/or by-products. Most of the waste obtained from chickpea milling includes the coat (or hull). Hull from *kabuli* chickpeas is an excellent source of dietary fibre and bioactive compounds, mostly phenolics [15], while hull from similarly black-coated pulses, such as black beans, is also rich of anthocyanins [16]. Based on these considerations, it would be interesting to evaluate the reuse of chickpea hulls as an innovative ingredient to fortify GF fresh pasta, also to meet the requirements of circular economy.

To the best of our knowledge, no studies have been done so far to reuse chickpea hulls in the production of food. Therefore, the aim of this work has been to evaluate the feasibility of using chickpea hulls obtained from *kabuli* and *Apulian black* chickpeas as innovative ingredients in the production of potentially functional GF fresh pasta, *orecchiette*-shaped. Hull-fortified pasta was characterised in comparison with control GF fresh pasta in terms of nutritional characteristics, bioactive compounds, cooking performance, colour, textural, and sensory liking.

## 2. Results and Discussion

### 2.1. Characteristics of Chickpea Hulls

Proximate composition, content of bioactive and anti-nutritional compounds, antioxidant activity, and colour parameters of chickpea hulls are reported in Table 1. Significant differences (*p* < 0.05) were found between Apulian black (ABH) and kabuli hull (KH) for all the parameters analysed.

ABH had higher proteins, lipids, and carbohydrates than KH, which was characterised by the highest dietary fibre and ash content. Furthermore, ABH showed the highest levels of anti-nutritional compounds, including verbascose, which was not detected in KH. ABH also showed a significantly higher content of total phenolics, anthocyanins, and carotenoids, which are recognized as antioxidants. Therefore, ABH showed a higher antioxidant activity than KH. The colour of the seed coat is attributed to polyphenols, mainly flavonoids [17]. Indeed, it has been reported that pigmented seeds contain higher levels of anthocyanins [18], as was found in ABH, which appeared dark brown (Figure 1). Instead, KH was characterised by the highest values of *a** and *b**.

Chickpea hull, like the external coat of other legumes, is known for its high dietary fibre content. Zhong et al. [19] reported fibre contents between 74.90 and 84.20 g/100 g. KH was within this range, while ABH was below the minimum value, and also below the value of 70.37 g/100 g reported by Niño-Medina et al. [15]. Dietary fibre contributes to human health, reducing the incidence of chronic diseases like cancer, and modulating intestinal mobility, cholesterolemia, and glucose homeostasis [20].

Legumes contain several anti-nutritional factors, including phytic acid and undigestible oligosaccharides. Phytates reduce the bio-availability of several important divalent cations (e.g., Fe, Zn, Ca, and Mg), hampering their absorption by the small intestine [21]. The raffinose family oligosaccharides (RFO), or α-galactosides, are a class of oligosaccharides, including raffinose, verbascose, and stachyose, which have a α (1→6) linkage between sucrose and the galactosyl subunit [22]. Due to the lack of α-galactosidase, humans do not digest RFO, which are metabolized by intestinal microflora producing carbon dioxide, hydrogen, and small quantities of methane [23]. At the same time, since RFO act as substrate for intestinal bacteria, they are also considered as prebiotics [24]. In the food application, however, RFO are mostly considered as antinutrients, because they cause flatulence and intestinal discomfort [25]. In particular, RFO reduce the acceptability of chickpea seeds [26].

In addition, the dehulling step carried out during legume milling can influence the distribution of these compounds between primary products and by-products. Generally, dehulling leads to an increase in the concentration of anti-nutritional compounds within the cotyledon fraction that constitutes the flour, rather than seed hulls [27]. Therefore, the content of oligosaccharides and phytates observed in ABH and KH was lower than those reported by other researchers in whole chickpeas [10,28].

Sucrose, identified as the predominant soluble sugar in chickpea seeds [29], is not an antinutrient, but it is involved in the formation of the α(1-6) linkage with galactosyl subunits that form the RFO structure. Furthermore, a positive correlation has been reported in chickpea seeds between RFO and some of the initial substrates of RFO biosynthetic pathway, such as myo-inositol and sucrose [29,30].

As for bioactives, other researchers reported that hulls isolated from pigmented chickpeas contain a higher amount of polyphenols and flavonoids and exhibit higher antioxidant activity than non-pigmented chickpea varieties [31]. Polyphenols and flavonoids can prevent or reduce lipid oxidation and scavenge free oxygen radicals [32]. A high content of fibre and bioactive compounds, coupled with a low content of anti-nutritional compounds, make chickpea hull, especially ABH, a potential ingredient to develop new functional food products.

### 2.2. Nutritional Composition of GF Fresh Pasta

Table 2 reports the nutritional characteristics of fresh GF pasta, orecchiette-shaped. Significant differences (*p* < 0.05) were found among enriched samples and the control for all parameters. Specifically, ABH-enriched pasta was characterised by a higher protein, lipid, and moisture content than both KH-enriched pasta and control. KH-enriched pasta had the highest fibre and ash contents, followed by the ABH-enriched one and control. Moreover, no significant difference (*p* > 0.05) was found for carbohydrate and protein contents between KH-enriched pasta and the other samples. By reducing carbohydrates and increasing fibres, both types of hull led to a reduction of the energy value.

In accordance with our results, Kaya et al. [33] reported a reduction of protein content in Turkish noodles enriched with pea and faba bean hull, as the level of hull increased.

Legume hulls can be used as a source of fibre in different food products, such as Turkish noodles and meat products [33,34]. Noodles substituted with different levels (2.5, 5, and 10%) of hull obtained from different legumes (green lentil, red lentil, faba bean, and pea) showed a significantly higher total dietary fibre content than control [33]. In particular, the addition of legume hull increased the fibre content of noodles up to 8.3% at the maximum level of substitution (10%) [33]. The same tendency was observed by other researchers in chicken nuggets added of pea and chickpea hull [34].

In accordance with these results, the addition of 8% ABH and KH improved the fibre content of GF pasta, reaching the value (6 g/100 g fresh product) required to claim “high fibre” in the label [35]. It has to be considered that in a study aimed at developing a high-fibre wheat-based bread, a very high level (21%) of broad bean hull was required to achieve a fibre content similar to that of KH-enriched pasta [36]. Our results, instead, showed the possibility to improve the nutritional characteristics of GF fresh pasta with a lower amount of KH or ABH due to their high fibre content, compared to other legumes.

### 2.3. Cooking Properties and Firmness of GF Fresh Pasta

Significant differences (*p* < 0.05) were found between enriched pasta and control for cooking properties, especially cooking loss and firmness. Considering the cooking loss (Figure 2a), both ABH and KH hull improved the quality of GF pasta. ABH, in particular, reduced the cooking loss from 4.98 to 3.49%. Firmness, instead, was higher in KH-enriched pasta (Figure 2b). ABH-enriched pasta had a higher water absorption than both KH-enriched pasta and control, which were characterised by a similar value (Figure 2c).

Low cooking loss, high firmness, and ability to keep surface integrity are parameters related to good cooking quality of pasta [8]. Cooking loss is defined as the amount of solids dissolved in water during cooking and is an indicator of structural integrity [37]. The cooking loss of good quality pasta is below 10% [8]. All samples examined showed a cooking loss much below 10%, therefore their quality was good. Moreover, cooking loss was lower, and firmness higher, than those observed by Kaya et al. [33] in Turkish noodles enriched with legume hull, indicating that we selected as control a good formulation, which further improved by adding hull.

Kaya et al. [33] reported that incorporation of green lentil, red lentil, faba bean, and pea hulls in the noodles at 2.5, 5, or 10% had no significant effect on both cooking loss and hardness, except for pasta added with faba bean hull at 10%, where a significant increase in hardness was found compared with control. On the contrary, Kaur et al. [38] observed an increase of cooking loss in pasta fortified with bran of different cereals, such as wheat, rice, barley, and oat. In agreement with these results, Brennan et al. [39] found higher cooking loss and lower firmness in pasta added of various types of fibre (HiSol, pea fibre or bamboo fibre), compared with control. In contrast with our and Kaya et al.’s [33] results, cooking loss increased at increasing level of dietary fibre added, whereas firmness generally decreased [39]. On the other hand, the use of soluble, gel-forming, high molecular weight fibres like xanthan, guar, and locust bean gum, allows to reduce cooking loss and increase firmness. These hydrocolloids, indeed, are able to form a network around the starch granules, restricting excessive swelling and diffusion of amylose [39]. This may explain our results, where all types of pasta were characterised by low cooking losses. However, in contrast with evidence reported in the literature, in our study, both ABH and KH chickpea hulls improved the cooking performance of pasta, reducing cooking loss and increasing firmness. The reduction of cooking loss induced by these ingredients could be linked to their high firmness. However, the high water absorption found in ABH-enriched pasta may have affected the firmness of this product, which was lower than in KH-enriched pasta.

The quality of pasta is influenced by both the type and amount of added fibre [40]. Soluble fibre, for example, is lost during cooking, increasing the cooking loss; therefore, its addition negatively affects the technological quality of pasta [41]. Chickpea, instead, is characterised by a marked prevalence of insoluble over soluble fibre [42]. Regarding the effect of the amount, oat bran fibre (mainly insoluble) had a different effect on cooking loss depending on the level of addition, with unvaried or lower cooking loss at a bran level of up to 5%, while similar or lower cooking loss was observed at values higher than 7.5% [41]. This result is probably due to the fact that, at low concentrations, the fibre may be dispersed and incorporated into the protein—starch matrix. At higher degrees of substitution, disruptions in the protein matrix by an oat bran particle become important and would promote water absorption and facilitate starch granule swelling and rupture [41]. These results paralleled the findings of Manthey et al. [43] in a study where wheat bran was considered.

As for water absorption, it is closely related to the cooking quality and consistency of pasta [44]. Insufficient water absorption could result in hard texture. On the opposite, excessively high water absorption may produce very soft and sticky pasta. Significant increases in water absorption have been already reported in noodles added of fibre, such as black or purple wheat bran [45].

### 2.4. Characteristics of Raw and Cooked GF Fresh Pasta

The content of bioactive and anti-nutritional compounds, antioxidant activity, and colour parameters of GF fresh pasta are reported in Table 3. All parameters were significantly influenced (*p* < 0.05) by fortification (F) and cooking (K) factors and their interaction fortification × cooking (F × K) (Table 4), except for total phytates and *b**. Total phytates and *b**, indeed, were not influenced by interaction F × K and the factor cooking. Furthermore, significant differences (*p* < 0.05) were found among samples for all parameters, both before and after cooking, except for verbascose and stachyose. Despite their presence in the hull, these anti-nutritional compounds were not present in pasta.

As for the nutritional compounds, both raw and cooked KH- and ABH-enriched pasta showed lower contents of total phytates than control. The reduction of phytates might be due to a dilution effect in the matrix of pasta, mostly constituted of refined rice flour. Polished rice flour is known to have the lowest amount of phytates among all cereals [46].

Raffinose was found both in raw and cooked ABH-enriched pasta, which was also characterised by the highest content of sucrose, followed by KH-enriched pasta and control. These anti-nutritional compounds showed a different beahviour with cooking.

No significant difference was found in the phytate content by comparing the raw and cooked samples, probably due to the heat-stable characteristics of phytic acid [47], whereas raffinose and sucrose significantly decreased. In accordance with our results, Arribas et al. [48] found a slight, not statistically significant, increase of total inositol content in cooked rice/bean-based pasta supplemented with carob. A significant increase in phytic acid was found by other researchers in fresh pasta with fermented whole wheat semolina [49]. The increase could be due to the formation of complexes among inositol and other food components (e.g., minerals, proteins, or starch) during the production of pasta, followed by a release during cooking [50]. Furthermore, the increase could be also due to a concentration effect because some soluble components, such as sugars or phenols, are lost in the cooking water, decreasing the total weight of pasta [48]. The reduction of raffinose and sucrose content during cooking, indeed, could be due to leaching in the cooking water or to thermal hydrolysis. A decrease of oligosaccharides of the raffinose family was reported also by Laleg et al. [51] in cooked legume-based pasta.

Considering the bioactive compounds, ABH-enriched pasta showed the highest total phenolic compounds, total anthocyanins, and total carotenoids, followed by KH-enriched pasta and control. Both types of fortified pasta showed a higher antioxidant activity than control.

However, the cooking process induced a decrease of these compounds and antioxidant activity compared to uncooked pasta, especially in KH-enriched samples. Moreover, KH-enriched pasta lost the total anthocyanins, which were present only in ABH-enriched cooked pasta. The reduction of total bioactive compounds may be explained by the fact that many of these compounds are hydrophilic molecules, such as phenolic compounds, which could be leached into the cooking water [8]. Other studies show that the addition of Apulian black chickpea induced an improvement of the nutritional characteristics of bakery products and canned sterilized purée [12,13]. Summo et al. [13] reported higher values of total anthocyanins, total carotenoids, total phenolics, and antioxidant activity in canned sterilized purées obtained from Apulian black chickpea flour than in purées obtained from kabuli chickpea.

The presence of anthocyanins and carotenoids in both chickpea hulls, especially in ABH, influenced pasta colour. Therefore, considering the uncooked pasta, ABH- and KH-enriched samples had higher *a** and *b** values than control, with ABH-enriched pasta also having the lowest *L** value. Despite the fact that the pigments were reduced, the colour parameters of fortified pasta increased with cooking, except for *b** in ABH-enriched samples. A reduction in the brightness of noodles enriched of legume bran has been previously reported [33].

### 2.5. Sensory Liking

The sensory liking (ranking test) of fortified and control pasta, effected by a semi-trained panel, is reported in Table 5. No significant differences (*p* > 0.05) were found among the samples for the attributes analysed, except for “aftertaste”. Aftertaste of control pasta was the least accepted, while the KH-enriched pasta was the most acceptable for this attribute. The aftertaste of ABH-enriched pasta was similar to both KH-enriched pasta and control.

In general, KH-enriched pasta was more appreciated than the ABH-enriched one, except for the “appearance”. Although not significantly, “appearance” and “colour” of enriched samples were scored higher than control, indicating that the presence of dark brown/yellowish hull particles on pasta surface was positively appreciated. On the contrary, Kaya et al. [33] indicated that the appearance of semolina-based noodles enriched of legume hull was little appreciated, with a progressive worsening as the level of substitution increased. Only the pea hull addition was appreciated, due to its light and creamy colour. The difference with our results was probably due to the fact that the GF control pasta, being rice-based, was very pale, almost white. The addition of hull improved colour, intensifying it and making it more similar to that of conventional durum wheat pasta, which remains a standard reference for good quality pasta. More specifically, the addition of KH made the product become particularly similar to semolina pasta, whereas the addition of ABH made it more similar to pasta obtained from wholemeal semolina.

Considering “smell” and “taste” attributes, the fortification with KH was the most appreciated. KH-enriched pasta was the most preferred also for “texture”, which was similar to the control. Overall, the sensory evaluation revealed that the addition of 8% KH and ABH to GF fresh pasta resulted in acceptable pasta properties, as also reported by Levent [52] for GF *tarhana* added of legume hulls.

## 3. Materials and Methods

### 3.1. Materials

*Kabuli* (KH) and *Apulian black* (ABH) chickpea hulls were supplied by Andriani S.p.A. (Gravina in Puglia, Italy) and A.R.T.E. Agricola s.r.l. (Candela, Italy), respectively. Rice flour (0.5 g/100 g fats, 82 g/100 g carbohydrates, 0.5 g/100 g fibres, 7 g/100 g proteins, and 0.1 g/100 g ash on fresh matter) was purchased at a local retailer. Xanthan gum and carob seed flour were purchased from Special ingredients Ltd. (Chesterfield, UK) and Farmalabor (Canosa di Puglia, Italy), respectively.

### 3.2. Fresh Pasta Production

Pasta dough was obtained according with the method reported by [8] with some modifications consisting in replacing the psyllium with other additives. Briefly, rice flour and a freshly prepared gel (2.5 g xanthan gum and 2.5 g carob seed flour were dissolved in 100 g of tap water at room temperature) were manually mixed in a bowl in a 50:50 ratio to obtain the control dough. To prepare hull-enriched pasta, KH or ABH (8% *w*/*w*) were also incorporated in the dough. The quantity of hull added was previously set as the maximum possible which still allowed the dough to be kneaded properly. After 20 min of manual kneading, the dough was left to rest for 15 min at room temperature, then it was hand-shaped into a traditional short-cut pasta type produced in the Apulia region (Southern Italy), called *orecchiette* (literally “little ears”), following the procedure described by [4]. This kind of pasta has a concave circular shape, diameter of 2.0–2.5 cm, and thickness of 2–3 mm. After shaping, pasta was put on a wooden vessel, covered with a cotton cloth, and left to dry at room temperature until values of 28.00 ± 2.0 g/100 g moisture and 0.95 ± 0.02 a_w_ were reached, accomplishing the legal requirements for fresh pasta [2]. Pasta samples were then lyophilized (Lyovapor L-200, Buchi Italia s.r.l., Cornaredo, Italy), and ground (HM-5735, Hoomei Electrical Appliance Co., Monza, Italy) for subsequent analyses, except for the determination of cooking properties, colour, and texture.

### 3.3. Determination of Proximate Composition

Protein (total nitrogen × 6.25), ash, and moisture content were determined according to the AOAC methods 979.09, 923.03, and 925.10, respectively [53]. Total dietary fibre content was determined by an enzymatic–gravimetric procedure according to the AOAC Official Method 991.43 [53]. The lipid content was determined by the SER 148 automated extractor (Velp Scientifica, Usmate, Italy) using diethyl ether (Sigma Aldrich, Milan, Italy) as extracting solvent, as described in the AOAC Official Method 945.38 F [53]. The carbohydrate content was determined as difference. The determinations were carried out in triplicate.

### 3.4. Determination of Bioactive Compounds

The total carotenoid content was determined using the method reported by Pasqualone et al. [54] and a calibration curve was prepared by different concentrations of β-carotene (Sigma-Aldrich Chemical Co., St. Louis, MO, USA) in order to express the carotenoid content in the sample as mg/kg of β-carotene on dry matter.

The total anthocyanin content was determined using the method reported by [54] and was expressed as mg/kg of cyanidin 3-*O*-glucoside. A calibration curve was prepared by using a different concentration of the cyanidin 3-*O*-glucoside standard (Phytoplan, Heidelberg, Germany).

The content of total phenolic compounds (TPC) and the antioxidant activity (AA) were assessed following the method proposed by Pasqualone et al. [55]. In order to express the total phenolic compounds content as mg/g of ferulic acid on dry matter, a calibration curve with ferulic acid at different concentrations was prepared. The antioxidant activity was evaluated by the 2,2-diphenyl-1-picrylhydrazyl (DPPH) radical scavenging capacity assay and expressed as μmol 6-hydroxy-2,5,7,8-tetramethylchroman-2-carboxylic acid (Trolox) equivalent/g on dry matter.

All determinations were carried out in triplicate.

### 3.5. Determination of Total Phytates

The content of total phytates was measured according to the method reported by Summo et al. [10]. The absorbance of extracts at 500 nm was considered. In order to express the phytates content in the sample as mg/g of phytic acid on dry matter, a calibration curve was prepared by different concentrations of phytic acid (Sigma-Aldrich Chemical Co., St. Louis, MO, USA) and the results were multiplied by 0.282 (molar ratio of phytate–phosphorus in a molecule of phytate). The analysis was carried out in triplicate.

### 3.6. Determination of Raffinose Family Oligosaccharides and Soluble Sugars

Raffinose family oligosaccharides (verbascose, stachyose, raffinose) and soluble sugars (sucrose) were determined by high-performance liquid chromatography (HPLC) (Agilent Technologies, Santa Clara, CA, USA), equipped with Refractive Index Detector (RID 1260, Agilent Technologies, Santa Clara, CA, USA), as previously reported in [56], with slight modifications. The HPLC separation was carried out isocratically at a 0.8-mL/min flow rate through a 300 × 7.8 mm cation exchange column (Rezex RCM column, Ca^2+^, 8 μm, Torrance, CA, USA) maintained at 80 °C. Deionized water was used as mobile phase. The identification was carried out comparing the retention time with that of the corresponding standard (Merck KGaA, Darmstadt, Germany). A calibration curve for each oligosaccharide was prepared for the quantification. The analysis was carried out in triplicate and the results were expressed as mg/g of oligosaccharide on dry matter.

### 3.7. Colour Determination

Colour indices of fresh pasta (*L**, *a** and *b**) were determined by means of the reflectance colorimeter Chroma Meter CR-300 (Konica Minolta Sensing, Osaka, Japan), with 10° Standard Observer and D65 illuminant.

### 3.8. Cooking Properties

Pasta was cooked in boiling distilled water at 1:10 (*w*/*v*) pasta to water ratio, without the addition of salt, as reported by Pasqualone et al. [4]. Optimum cooking time (OCT) was determined preliminary by removing two pasta pieces from the boiling water at 30 s intervals and cutting them to check the white and opaque core gradually disappearing, according to the AACC method 16–50 [57]. After cooking and draining, samples were rinsed with distilled water and allowed to rest for 5 min. Cooking loss was evaluated by combining the cooking and rinse waters, measuring total volume, putting 20 mL in a tarred Petri dish, and evaporating to dryness in an air-oven at 105 °C until a constant weight was reached [58]. The residue, scaled up to total volume, was expressed as a percentage of the original pasta sample weight. The water absorption (A) of pasta was determined as:A = (W − W0)/W0 × 100
where W and W0 were the weight of cooked and raw pasta, respectively. Cooking tests and subsequent determinations were performed in duplicate.

### 3.9. Firmness Determination

Firmness of cooked fresh pasta samples was carried out using a Z1.0 TN texture analyzer (Zwick Roell, Ulm, Germany) equipped with a blade set with guillotine. The test was conducted on the single pasta pieces, as described by the official method 66–50.01 [57], in the following conditions: 1.0 mm/s test speed; 50 N load cell. Firmness value was considered as the maximum cut force and expressed in Newton (N). Data acquisition was performed using the TestXPertII v3.41 software (Zwick Roell, Ulm, Germany). Pasta pieces were randomly selected and the mean of ten replicates was considered.

### 3.10. Sensory Liking

Seventeen semi-trained panelists were recruited based on their previous experience in the sensory evaluation of fresh pasta among technicians and researchers of the laboratory of the Food Science and Technology unit of the Department of Plant, Soil, and Food Sciences of the University of Bari, Italy. All panel members had neither food allergies nor intolerances and were regular consumers of fresh pasta, rice, and chickpeas. Pre-test sessions were carried out: (i) to define the procedure to prepare the samples; (ii) to define the method to compare the samples following their liking degree for each attribute according to a ranking test [59]. The study protocol followed the ethical guidelines of the laboratory. Panelists were given information about study aims, and individually written informed consent was obtained from each participant. All tested samples were food-grade. The samples, coded with random numbers, were cooked in unsalted boiling water at their OCT. After cooking and draining, each type of pasta was briefly rinsed under tap water. Participants were asked to evaluate and arrange the pasta samples on a scale from 1 (the least liked sample) to 3 (the most liked sample) according to their degree of liking for six sensory attributes—appearance, colour, typical pasta smell, texture, taste, and aftertaste, using an evaluation form. The obtained data were reported as the sum of ranks for each sample.

### 3.11. Statistical Analysis

The experimental data of chickpea hulls (ABH and KH) and GF fresh pasta were subjected to one-way ANOVA, followed by Tukey’s HSD test. The two-way ANOVA analysis was carried out considering the type of pasta sample and physical state (raw and cooked) as factors. Significant differences among the values of all parameters were determined at *p* ≤ 0.05 by the Minitab 17 Statistical Software (Minitab, Inc, State College, PA, USA, 2010). Sensory data were analysed by the Friedman test using the XLStat software (Addinsoft SARL, New York, NY, USA). *Post hoc* analysis with Wilcoxon signed-rank test was conducted by applying the Bonferroni correction, resulting in a significance level set at *p* < 0.05.

## 4. Conclusions

A high content of fibre and bioactive compounds, coupled with a low content of anti-nutritional compounds, make chickpea hull, especially ABH, a suitable ingredient to develop new functional food products. Our results showed, indeed, the possibility to improve the nutritional characteristics of GF fresh pasta with a low amount of both *Apulian black* and *kabuli* chickpea hulls. Furthermore, the enrichment of pasta positively influenced the colour of the end product and improved the firmness and cooking performance. 

The functionalization of hand-made *orecchiette*-shaped pasta innovated a traditional food product and increased its healthy value. Moreover, this strategy would add value to by-products of the chickpea milling process and help increase the antioxidant dietary intake, especially with the *Apulian black* chickpea, which was characterised by the highest bioactive compounds.

On this basis, promoting the use of chickpea hull in food production could improve the nutritional characteristics and physical characteristics of food products, increasing the consumption of fibre and bioactive compounds having antioxidant activity.

## Figures and Tables

**Figure 1 molecules-26-04442-f001:**
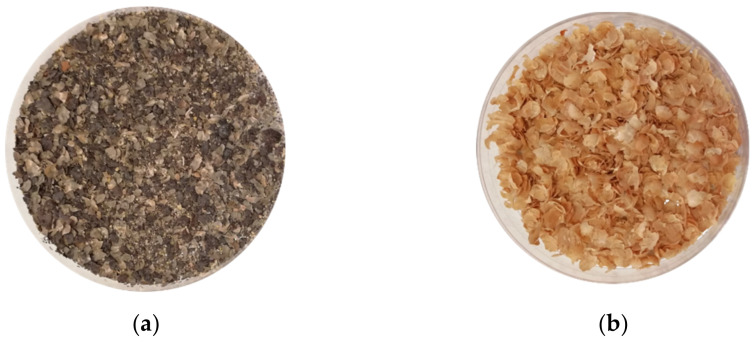
Colour of chickpea hull: (**a**) dark brown *Apulian black* chickpea hull; (**b**) yellowish *kabuli* chickpea hull.

**Figure 2 molecules-26-04442-f002:**
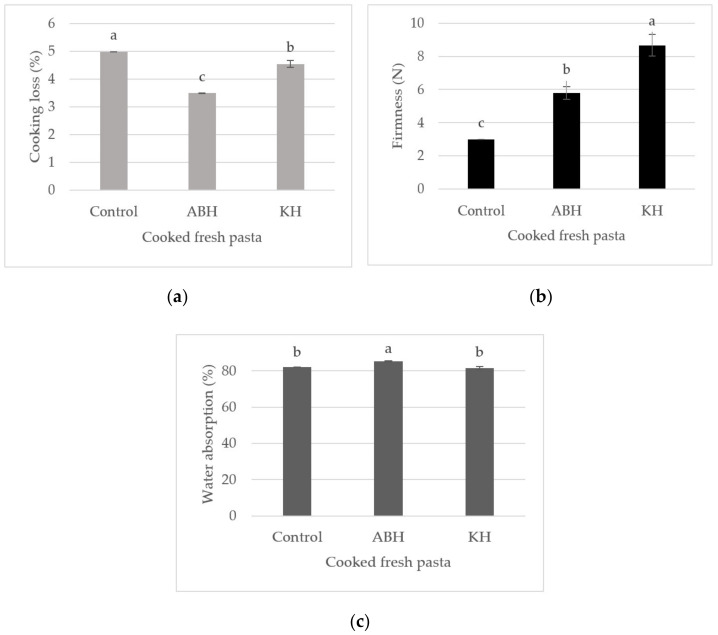
Cooking loss (**a**), firmness (**b**), and water absorption (**c**) of cooked GF fresh pasta enriched with 8 g/100 g of *Apulian black* (ABH) or *kabuli* (KH) chickpea hull, compared to control pasta. Different letters indicate significant differences at *p* < 0.05.

**Table 1 molecules-26-04442-t001:** Proximate composition, anti-nutritional compounds, bioactive compounds, antioxidant activity, and colour characteristics of *Apulian black* (ABH) and *kabuli* (KH) chickpea hull used in the formulation of GF fresh pasta (mean value ± standard deviation; one-way ANOVA was applied).

Parameter	ABH	KH
Proximate composition		
Carbohydrates (g/100 g d.m.)	16.30 ± 0.70 ^a^	1.31 ± 0.68 ^b^
Total dietary fibres (g/100 g d.m.)	65.26 ± 0.69 ^b^	82.75 ± 0.67 ^a^
Proteins (g/100 g d.m.)	11.26 ± 0.12 ^a^	7.85 ± 0.08 ^b^
Lipids (g/100 g d.m.)	2.48 ± 0.08 ^a^	1.22 ± 0.09 ^b^
Ashes (g/100 g d.m.)	4.70 ± 0.04 ^b^	6.87 ± 0.02 ^a^
Anti-nutritional compounds		
Total phytates (mg/g phytic acid d.m.)	3.82 ± 0.28 ^a^	1.83 ± 0.01 ^b^
Verbascose (mg/g d.m.)	10.48 ± 0.55	^1^ n.d.
Stachyose (mg/g d.m.)	14.24 ± 0.88 ^a^	7.35 ± 0.16 ^b^
Raffinose (mg/g d.m.)	25.44 ± 0.38 ^a^	9.94 ± 0.76 ^b^
Soluble sugars		
Sucrose (mg/g d.m.)	18.5 ± 0.34 ^a^	9.4 ± 0.50 ^b^
Bioactive compounds		
Total phenolic compounds (mg/g ferulic acid d.m.)	0.81 ± 0.01 ^a^	0.63 ± 0.03 ^b^
Total anthocyanins (mg/kg cyanidin 3-*O*-glucoside d.m.)	225.64 ± 0.52 ^a^	64.77 ± 0.14 ^b^
Total carotenoids (mg/kg β-carotene d.m.)	62.16 ± 0.10 ^a^	25.01 ± 0.59 ^b^
Antioxidant activity (µmol/g Trolox ^2^ d.m.)	4.91 ± 0.17 ^a^	3.34 ± 0.09 ^b^
Colour characteristics		
*L**	51.07 ± 0.04 ^b^	67.90 ± 0.05 ^a^
*a**	0.62 ± 0.03 ^b^	4.69 ± 0.01 ^a^
*b**	9.19 ± 0.05 ^b^	21.12 ± 0.02 ^a^

^1^ n.d. = not detected; ^2^ Trolox = 6-hydroxy-2,5,7,8-tetramethylchroman-2-carboxylic acid. Different letters in row indicate significant differences (*p* < 0.05).

**Table 2 molecules-26-04442-t002:** Nutritional characteristics of GF fresh pasta enriched with 8 g/100g of *Apulian black* (ABH) or *kabuli* (KH) chickpea hull, compared to control pasta. All values (mean value ± standard deviation) are expressed on fresh weight basis. One-way ANOVA was applied.

Nutritional Parameter	Type of Pasta		
	Control	ABH-Enriched	KH-Enriched
Carbohydrates (g/100 g)	59.80 ± 0.19 ^a^	52.35 ± 0.23 ^b^	52.89 ± 0.28 ^b^
Proteins (g/100 g)	5.23 ± 0.07 ^b^	5.56 ± 0.06 ^a^	5.22 ± 0.01 ^b^
Lipids (g/100 g)	0.42 ± 0.01 ^c^	0.51 ± 0.02 ^a^	0.46 ± 0.01 ^b^
Total dietary fibres (g/100 g)	5.75 ± 0.26 ^c^	10.42 ± 0.20 ^b^	13.44 ± 0.28 ^a^
Ashes (g/100 g)	0.62 ± 0.01 ^c^	0.92 ± 0.01 ^b^	1.24 ± 0.02 ^a^
Moisture (g/100g)	28.17 ± 0.20 ^b^	30.23 ± 0.09 ^a^	26.75 ± 0.11 ^c^
Energy value (kcal/100 g)	275.43 ± 0.52 ^a^	257.09 ± 0.44 ^c^	263.50 ± 0.70 ^b^

Different letters in row indicate significant differences (*p* < 0.05).

**Table 3 molecules-26-04442-t003:** Anti-nutritional and bioactive compounds, antioxidant activity, and colour parameters of raw and cooked GF fresh pasta enriched with 8 g/100 g of *Apulian black* (ABH) and *kabuli* (KH) chickpea hull, compared to control pasta (mean value ± standard deviation; two-way ANOVA was applied).

Parameters	Raw Fresh Pasta			Cooked Fresh Pasta		
	Control	ABH	KH	Control	ABH	KH
Anti-nutritional compounds						
Total phytates (mg/g phytic acid d.m.)	1.79 ± 0.04 ^ab^	1.31 ± 0.06 ^d^	1.41 ± 0.05 ^cd^	1.87 ± 0.14 ^a^	1.54 ± 0.17 ^bcd^	1.62 ± 0.04 ^abc^
Verbascose (mg/g d.m.)	^1^ n.d.	n.d.	n.d.	n.d.	n.d.	n.d.
Stachyose (mg/g d.m.)	n.d.	n.d.	n.d.	n.d.	n.d.	n.d.
Raffinose (mg/g d.m.)	n.d.	8.39 ± 0.08 ^a^	n.d.	n.d.	6.09 ± 0.37 ^b^	n.d.
Sucrose (mg/g d.m.)	7.52 ± 0.2 ^d^	16.08 ± 0.68 ^a^	9.02 ± 0.76 ^c^	4.42 ± 0.28 ^e^	11.75 ± 0.76 ^b^	7.69 ± 0.07 ^cd^
Bioactive compounds						
Total phenolic compounds (mg/g ferulic acid d.m.)	0.29 ± 0.01 ^b^	0.33 ± 0.02 ^a^	0.31 ± 0.01 ^ab^	0.14 ± 0.01 ^e^	0.24 ± 0.02 ^c^	0.18 ± 0.01 ^d^
Total anthocyanins (mg/kg cyanidin 3-*O*-glucoside d.m.)	n.d.	33.37 ± 1.20 ^a^	2.87 ± 0.08 ^c^	n.d.	20.59 ± 0.11 ^b^	n.d.
Total carotenoids (mg/kg β-carotene d.m.)	0.66 ± 0.00 ^e^	8.24 ± 0.04 ^a^	1.97 ± 0.04 ^c^	0.43 ± 0.02 ^f^	5.86 ± 0.05 ^b^	1.71 ± 0.05 ^d^
Antioxidant activity (µmol/g Trolox ^2^ d.m.)	1.48 ± 0.06 ^c^	2.15 ± 0.06 ^a^	2.06 ± 0.02 ^ab^	1.24 ± 0.05 ^d^	1.92 ± 0.08 ^b^	1.44 ± 0.11 ^c^
Colour characteristics						
*L**	90.49 ± 0.84 ^a^	69.43 ± 1.60 ^d^	85.19 ± 0.62 ^b^	80.80 ± 1.17 ^c^	47.67 ± 0.51 ^e^	70.48 ± 0.77 ^d^
*a**	−0.16 ± 0.12 ^c^	2.23 ± 0.11 ^b^	2.02 ± 0.30 ^b^	−1.04 ± 0.12 ^d^	3.14 ± 0.33 ^a^	2.86 ± 0.09 ^a^
*b**	8.61 ± 0.42 ^c^	12.16 ± 0.51 ^b^	13.03 ± 0.86 ^b^	8.98 ± 0.64 ^c^	6.60 ± 0.39 ^d^	17.51 ± 0.78 ^a^

^1^ n.d. = not detected; ^2^ Trolox = 6-hydroxy-2,5,7,8-tetramethylchroman-2-carboxylic acid. Different letters in row indicate significant differences (*p* < 0.05).

**Table 4 molecules-26-04442-t004:** Statistic influence, analysed by two-way ANOVA, of *fortification* (F) and *cooking* (K) factors and their interaction (K × F) on anti-nutritional and bioactive compounds, and colour characteristics of GF fresh pasta. Fortification consisted in the addition of *Apulian black* or *kabuli* chickpea hull.

Parameters	*p*-Value		
	Fortification (F)	Cooking (K)	F × K
Anti-nutritional compounds			
Total phytates	*p* < 0.001	*p* < 0.01	*p* = 0.379
Verbascose	*p* < 0.001	*p* < 0.001	*p* < 0.05
Stachyose	*p* < 0.001	*p* < 0.001	*p* < 0.001
Raffinose	*p* < 0.001	*p* < 0.001	*p* < 0.001
Soluble sugars			
Sucrose	*p* < 0.001	*p* < 0.001	*p* < 0.001
Bioactive compounds			
Total phenolic compounds	*p* < 0.001	*p* < 0.001	*p* < 0.001
Total anthocyanins	*p* < 0.001	*p* < 0.001	*p* < 0.001
Total carotenoids	*p* < 0.001	*p* < 0.001	*p* < 0.001
Antioxidant activity	*p* < 0.001	*p* < 0.001	*p* < 0.001
Colour characteristics			
*L**	*p* < 0.001	*p* < 0.001	*p* < 0.001
*a**	*p* < 0.001	*p* < 0.01	*p* < 0.001
*b**	*p* < 0.001	*p* = 0.308	*p* < 0.001

**Table 5 molecules-26-04442-t005:** Sensory ranking test score (sums of 17 semi-trained consumers) of GF fresh pasta enriched with 8 g/100 g of *Apulian black* (ABH) or *kabuli* (KH) chickpea hulls, compared to control pasta.

	Type of Pasta		
Sensory Attribute	Control	ABH-Enriched	KH-Enriched
Appearance	28 ^a^	39 ^a^	35 ^a^
Colour	26 ^a^	37 ^a^	39 ^a^
Smell	27 ^a^	35 ^a^	40 ^a^
Texture	38 ^a^	27 ^a^	37 ^a^
Taste	30 ^a^	31 ^a^	41 ^a^
Aftertaste	26 ^b^	33 ^ab^	43 ^a^

Different letters in row indicate significant differences among the rank sums of 17 semi-trained consumers for each product analysed by the non-parametric Friedman test, followed by Wilcoxon signed-rank tests applying the Bonferroni correction, resulting in a significance level set at *p* < 0.05. A smaller sum of the rank indicates the worst like sensory attribute.

## Data Availability

The data presented in this study are available upon request from the corresponding author.

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
