# Peer review of "Kabuli and Apulian black Chickpea Milling By-Products as Innovative Ingredients to Provide High Levels of Dietary Fibre and Bioactive Compounds in Gluten-Free Fresh Pasta"

_molecules, 2021, doi:10.3390/molecules26154442_

Round 1

Reviewer 1 Report

I suggest deciding for American or British English spelling because sometimes you use British f.ex. “firbre”, “colour” and sometimes American f.ex. “characterized”, "color"

Line 88: “reported by [15]” it would be better to write reported by Niño-Medina et al. [15].

Table 1. I do not find it appropriate to name CIE L*a*b* parameters as “Brown index”, “Red index” and “Yellow index” usually it is described as L referring to the white-black, a*-green to red and b*blue – yellow, however, if you found in the literature the names “brow, red and yellow indexes” it is up to you whether you use it or not.

Line 111: “flavonoids can prevent or reduce lipid peroxidation” – I would rewrite as ‘’lipid oxidation

I think an overview of results in the tables would be better for readers if you mention a percentage of ABH and KH added to the fresh pasta (8%).

Line 130: “enriched of”… replace with “enriched with”

Line 137: “The same behaviour was observed” replace with f.ex. “The same tendency was observed”

Line 301: “Kaya et al. [27] report” replace with “reported”

Author Response

Responses to Reviewer 1

NOTE: All the suggestions have been implemented, thank you. All modifications made for Reviewer 1 are in red text.

I suggest deciding for American or British English spelling because sometimes you use British f.ex. “firbre”, “colour” and sometimes American f.ex. “characterized”, "color"

Response: Sorry for inaccuracy. British English has been used throughout the manuscript.

Line 88: “reported by [15]” it would be better to write reported by Niño-Medina et al. [15].

Response: Done (see line 88).

Table 1. I do not find it appropriate to name CIE L*a*b* parameters as “Brown index”, “Red index” and “Yellow index” usually it is described as L referring to the white-black, a*-green to red and b*blue – yellow, however, if you found in the literature the names “brow, red and yellow indexes” it is up to you whether you use it or not.

Response: CIE L*a*b* are commonly referred to as “brown index”,” yellow index” and “red index” in the field of semolina pasta-making (See references below). However we considered that in our work we do not deal with semolina pasta, but with an unconventional type of gluten-free pasta. Further, in the manuscript we referred to “brown index”,” yellow index” and “red index” also for colour data of chickpea bran.

Therefore, we deleted “brown index”,” yellow index” and “red index” everywhere (text and Tables) and mentioned only L*a*b*. Of course, the values of brown index, which were calculated as 100-L*, have now been changed directly to L*, therefore all figures have been changed. (See Table 1, Table 3 and line 402).

Yellow index, red index  and brown index are mentioned in:

Abecassis, J., Abbou, R., Chaurand, M., Morel, M.H. and Vernoux, P., 1994. and Pressure in the Extruder and on Pasta Quality. Cereal Chem, 71(3), pp.247-253.

Autran, J.C., Abecassis, J. and Feillet, P., 1986. Statistical evaluation of different technological and biochemical tests for quality assessment in durum wheats. Cereal Chem, 63(5), pp.390-394.

Taddei, F., Galassi, E., Nocente, F. and Gazza, L., 2021. Innovative Milling Processes to Improve the Technological and Nutritional Quality of Parboiled Brown Rice Pasta from Contrasting Amylose Content Cultivars. Foods, 10(6), p.1316.

Borrelli, G.M., De Leonardis, A.M., Fares, C., Platani, C. and Di Fonzo, N., 2003. Effects of modified processing conditions on oxidative properties of semolina dough and pasta. Cereal Chemistry, 80(2), pp.225-231.

Line 111: “flavonoids can prevent or reduce lipid peroxidation” – I would rewrite as ‘’lipid oxidation”

Response: Done (see line 122).

I think an overview of results in the tables would be better for readers if you mention a percentage of ABH and KH added to the fresh pasta (8%).

Response: Done. (see lines 137, 264, 303).

Line 130: “enriched of”… replace with “enriched with”

Response: Done (see line 142).

Line 137: “The same behaviour was observed” replace with f.ex. “The same tendency was observed”

Response: Done (see line 149).

Line 301: “Kaya et al. [27] report” replace with “reported”

Response: Done.

Reviewer 2 Report

This paper deals with valorisation of chickpea milling by-products through the production of gluten free pasta. Generally, the experiments were set up correctly and appropriate methods are used. However, the issue of anti-nutrients could be better elaborated.

Raffinose family oligosaccharides are usually considered anti-nutrients, although α-galactosides may have a beneficial effect by increasing the bifidobacteria population in the colon, so they are considered soluble fibre by some authors. I believe that the quality of the manuscript will be improved by critically assessing the role of these sugars. Also, in your work sucrose is also classified as an anti-nutrient. Can you explain that. In line 100 it is stated “The amount of anti-nutritional  compounds, polyphenols  and flavonoids shows a great intraspecific..” Are polyphenols and flavonoids antinutrients or not?

Author Response

Responses to Reviewer 2

NOTE: All the suggestions have been implemented, thank you. All modifications made for Reviewer 2 are in green text.

This paper deals with valorisation of chickpea milling by-products through the production of gluten free pasta. Generally, the experiments were set up correctly and appropriate methods are used. However, the issue of anti-nutrients could be better elaborated.

Raffinose family oligosaccharides are usually considered anti-nutrients, although α-galactosides may have a beneficial effect by increasing the bifidobacteria population in the colon, so they are considered soluble fibre by some authors. I believe that the quality of the manuscript will be improved by critically assessing the role of these sugars.

Response: Thanks for suggestion. A critical discussion of the role of raffinose family oligosaccharides (RFO) has been added (see lines 99-107).

Also, in your work sucrose is also classified as an anti-nutrient.

Response: We better explained that sucrose is not an antinutrient and added some information about its occurrence in chickpeas and its relation with RFO (see lines 114-118).

Can you explain that. In line 100 it is stated “The amount of anti-nutritional  compounds, polyphenols  and flavonoids shows a great intraspecific..” Are polyphenols and flavonoids antinutrients or not?

Response: Sorry for mistake, we deleted “polyphenols and flavonoidsin the sentence (it was a poorly written and therefore misleading sentence ).
